# Intake of Vitamin E and C in Women of Reproductive Age: Results from the Latin American Study of Nutrition and Health (ELANS)

**DOI:** 10.3390/nu13061954

**Published:** 2021-06-07

**Authors:** Dolores Busso, Andrea David, Reyna Penailillo, Guadalupe Echeverría, Attilio Rigotti, Irina Kovalskys, Georgina Gómez, Lilia Yadira Cortés Sanabria, Martha Cecilia Yépez García, Rossina G. Pareja, Marianella Herrera-Cuenca, Mauro Fisberg

**Affiliations:** 1Biomedical Research and Innovation Center, School of Medicine, Universidad de los Andes, Santiago 7550000, Chile; reyna.penailillo@gmail.com; 2Department of Nutrition, Diabetes and Metabolism, School of Medicine, Pontificia Universidad Católica de Chile, Santiago 8320000, Chile; andreadavidn@hotmail.com (A.D.); gecheverria@bio.puc.cl (G.E.); arigotti@med.puc.cl (A.R.); 3Center of Molecular Nutrition and Chronic Diseases, School of Medicine, Pontificia Universidad Católica de Chile, Santiago 8320000, Chile; 4Carrera de Nutriciόn, Facultad de Ciencias Médicas, Pontificia Universidad Catόlica Argentina, Buenos Aires C1059ABF, Argentina; ikovalskys@ilsi.org.ar; 5Departamento de Bioquímica, Escuela de Medicina, Universidad de Costa Rica, San José 94088, Costa Rica; georgina.gomez@ucr.ac.cr; 6Departamento de Nutrición y Bioquímica, Pontificia Universidad Javeriana, Bogotá 110111, Colombia; ycortes@javeriana.edu.co; 7Colegio de Ciencias de la Salud, Universidad San Francisco de Quito, Quito 171200841, Ecuador; myepez@usfq.edu.ec; 8Instituto de Investigación Nutricional, Lima 15026, Peru; rpareja@iin.sld.pe; 9Centro de Estudios del Desarrollo, Universidad Central de Venezuela (CENDES-UCV) and Fundación Bengoa, Caracas 1010, Venezuela; marianella.herrera@ucv.ve; 10Instituto Pensi, Fundação Jose Luiz Egydio Setubal, Hospital Infantil Sabara, São Paulo 04023062, Brazil; mfisberg.dped@epm.br; 11Centro de Excelencia em Nutrição e Dificuldades Alimentaes (CENDA) Instituto Pensi, Fundação José Luiz Egydio Setubal, Departamento de Pediatria, Universidade Federal de São Paulo, São Paulo 01239040, Brazil

**Keywords:** women of reproductive age, Latin America, intake, vitamin E, vitamin C

## Abstract

Vitamin E was identified as a lipophilic compound essential to maintain rat pregnancy. Low vitamin E intake during early pregnancy associates with congenital malformations and embryonic loss in animals and with miscarriage and intrauterine growth restriction in humans. Vitamin E protects cell membranes from lipoperoxidation and exerts non-antioxidant activities. Its function can be restored by vitamin C; thus, intake and circulating levels of both micronutrients are frequently analyzed together. Although substantial vitamin E inadequacy was reported worldwide, its consumption in Latin America (LatAm) is mostly unknown. Using data from the Latin American Study of Nutrition and Health (*Estudio Latinoamericano de Nutrición y Salud*, ELANS), we evaluated vitamin E and C intake in women of reproductive age (WRA) from eight LatAm countries and identified their main food sources. Two non-consecutive 24-h dietary recalls in 3704 women aged from 15 to 49 years and living in urban locations showed low average intake of vitamin E (7.9 mg/day vs. estimated average requirement (EAR) of 12 mg/day) and adequate overall vitamin C consumption (95.5 mg/day vs. EAR of 60 mg/day). The mean regional inadequacy was 89.6% for vitamin E and 36.3% for vitamin C. The primary food sources of vitamin E were fats and oils, as well as vegetables. Vitamin C intake was explained mainly by the consumption of fruit juices, fruits, and vegetables. Combined deficient intake of both vitamins was observed in 33.7% of LatAm women. Although the implications of low antioxidant vitamins’ consumption in WRA are still unclear, the combined deficient intake of both vitamins observed in one-third of ELANS participants underscores the need for further research on this topic.

## 1. Introduction

During the 1920s, three research groups reported almost simultaneously that laboratory rats consuming purified diets became sterile but rapidly recovered their fertility after green leaves or whole grains were added to their food [1,2,3]. These initial studies allowed the characterization and later purification of a dietary factor essential for reproduction in laboratory animals, which was later denominated vitamin E. 

Vitamin E is a mixture of eight different hydrophobic molecular forms: α, β, γ, and δ tocopherols and tocotrienols [4]. Only α-tocopherol is maintained at significant levels in circulating lipoproteins and body reserves such as the liver and adipose tissue. The retention of α-tocopherol is explained by the presence of two key liver enzymes: α-tocopherol transfer protein (α-TTP), which preferentially binds α-tocopherol and facilitates its secretion into plasma lipoproteins, and ω-hydroxylase, which catabolizes other forms for biliary and renal excretion [5]. 

Vitamin E functions as an antioxidant lipid that protects fatty acids composing phospholipids in the plasma membrane from oxidation by reactive oxygen molecules [6]. It also exerts non-antioxidant functions such as regulation of gene expression and intracellular cell signaling [7,8]. The antioxidant activity of vitamin E is strongly dependent on the action of ascorbic acid and other dietary components such as selenium and carotenoids [9]. In smokers, alpha-tocopherol disappearance from plasma correlates positively with oxidative stress and negatively with plasma ascorbic acid concentrations, suggesting that vitamins E and C have concomitant antioxidant functions [10]. Based on their joint physiological functions, circulating levels and daily intake of vitamin E and C are usually studied in parallel [11]. 

Some of the foods most enriched in vitamin E are vegetable oils, nuts, whole grains, and green leafy vegetables. Its Estimated Average Requirement (EAR) was set at 12 mg/day by the Institute of Medicine of the United States (US), based on studies demonstrating inhibition of hydrogen peroxide-induced erythrocyte hemolysis in men [11]. Vitamin E intake has been reported to be inadequate worldwide [12]. In the US and Spain, the proportion of women consuming α-tocopherol below EAR levels is as high as 98% and 82%, respectively [13,14]. Inadequate vitamin C intake (below 60 mg/day for women and 56 mg/day for 14–18-year-old-girls [11]) is not as dramatic as for vitamin E in most populations analyzed. Indeed, a recent study reported that the % of US subjects consuming less than the EAR for vitamin C in 2017–2018 was 47.4% [15]. 

Patients with genetic mutations in α-TTP or genes involved in lipoprotein metabolism (e.g., ApoB and MTTP), fat malabsorption syndromes, or hematologic disorders show neurological symptoms due to vitamin E deficiency [16]. By contrast, clinical symptoms in individuals consuming low vitamin E diets are rare and have only been detected in children with severe malnutrition [17]. However, higher vitamin E requirements and beneficial effects of supplementation have been reported in individuals with chronic diseases undergoing inflammation or oxidative stress, such as obesity [18], metabolic syndrome [19], or non-alcoholic fatty liver disease [20], suggesting that subclinical deficiency may also be relevant for human health. 

Although vitamin E was initially identified as a fertility factor, the precise molecular and cellular processes regulated by vitamin E during pregnancy are not entirely understood [17]. Alpha-TTP protein, the main intracellular vitamin E transporter, is present in first-trimester reproductive tissues such as decidua and trophoblasts in humans and rodents, and yolk sac in fish [21,22,23]. The highly conserved localization of this protein during the initial stages of pregnancy suggests that α-TTP may be involved in providing vitamin E to early embryos before the establishment of a functional placenta [24]. In line with this evidence, α-TTP knockout mice are infertile due to placental defects and embryonic demise at mid-gestation [25]. In animals as diverse as mice, poultry and zebrafish, deficient maternal vitamin E intake or provision to the embryo is associated with increased embryonic resorptions and congenital malformations [26,27,28]. In human pregnancy, studies in undernourished populations have associated low plasma α-tocopherol levels with higher incidences of miscarriage [29,30] and intrauterine growth restriction [31,32]. 

Different studies evaluating the efficacy of maternal supplementation with vitamin E on pregnancy outcomes related to oxidative stress, such as pre-eclampsia, preterm birth, neonatal death, and gestational diabetes mellitus have shown inconsistent and, overall, non-significant findings [33,34,35]. However, it must be pointed out that most clinical trials were performed in women with unknown plasma vitamin E status or intake during mid and late pregnancy. The protective effect of vitamin E from oxidative damage on the preimplantation embryo or early post-implantation conceptus cannot be ruled out because studies evaluating the effect of pregestational or first-trimester supplementation with this micronutrient have not been reported.

Current intake recommendations and reference intervals for biochemical markers of vitamin E status are the same in pregnant and non-pregnant women [36,37]. Expert panels from US and Europe concluded that the available data do not indicate an additional dietary vitamin E requirement during pregnancy based on the progressive rise of the α-tocopherol concentration in parallel with other lipids’ physiological increase during gestation [38] and the constant placental transfer of vitamin E reported during pregnancy [39]. Recent results from a cross-sectional analysis on 1003 pregnant US women (20–40 years old) from the 2001–2014 National Health and Nutrition Examination Survey (NHANES) showed that the percentages of women with usual intakes below the EAR was near 92% and 25% for vitamins E and C, respectively [40].

In this study, we sought to analyze the daily intake and food sources of vitamin E and vitamin C in Latin American (LatAm) women of childbearing age. We used a subsample of data obtained through the Latin American Nutrition and Health Survey (ELANS), which assessed intake in 3704 participants between 15 and 49 years old, living in urban settings from Argentina, Brazil, Chile, Perú, Colombia, Costa Rica, Ecuador, and Venezuela, using two non-consecutive 24-h dietary recalls. We determined the mean intake in the eight countries as a regional proxy, and in each country individually. To estimate the proportion of WRA with inadequate vitamin E and C levels, we compared individual intakes with EARs for each vitamin. We also correlated vitamin intake with different sociodemographic categories and identified local dietary sources of both micronutrients in each country. Our study showed a deficient vitamin E intake in WRA from LatAm and less dramatic but still high vitamin C inadequacy in some countries. A combined deficient intake of both antioxidant vitamins was found in one-third of LatAm WRA. 

## 2. Materials and Methods

### 2.1. Study Design 

ELANS is a household-based multi-national cross-sectional multicenter survey conducted from September 2014 to August 2015 in representative urban populations of eight Latin American countries: Argentina, Brazil, Chile, Colombia, Costa Rica, Ecuador, Perú, and Venezuela. Details on the ELANS study design and methodology have been published before Fisberg et al. [41]. Participants’ recruitment was performed through a random complex, multistage process stratified by urban geographical location, sex, age, and socioeconomic level (SEL), with a sample error of 3.49% at a 5% statistical significance level. From 9218 ELANS participants, only data from 3704 non-lactating non-pregnant women between 15–49.9 years of age were used for this study. 

### 2.2. Anthropometric Measurements

Anthropometric measurements were obtained by trained interviewers following standardized procedures. Bodyweight was measured with a calibrated electronic scale up to 200 kg with an accuracy of 0.1 kg after removing heavy clothing, pocket items, shoes, and socks. Height was measured with a portable stadiometer up to 205 cm with an accuracy of 0.1 cm. Body mass index (BMI; kg/m^2^) was derived from height and weight. Nutritional status was defined as underweight [BMI < 18.5 kg/m^2^], normal [BMI 18.5 to <25 kg/m^2^], overweight [BMI 25 to <30 kg/m^2^], and obese [BMI ≥ 30 kg/m^2^], for participants over 18 years of age and classified according to the cut-off criterion proposed by de Onis for World Health Organization (WHO) in 2007 [42,43].

### 2.3. Sociodemographic and Lifestyle Population Variables

The age on the interview date was registered and ELANS participants were classified into three age groups: 15–19, 20–34, and 35–49 years old. SELs were classified as high, medium, and low status using a questionnaire based on each country’s national indexes, as described in [41]. A detailed analysis of demographic and epidemiologic characteristics of WRA participating in ELANS has been published recently [44].

### 2.4. Dietary Assessment 

The 24-h dietary recall method was used as described elsewhere (Fisberg et al., 2016). Briefly, trained interviewers undertook two face-to-face household visits in two non-consecutive days, with a maximum interval of eight days between them, including both weekdays and weekends, to capture day-to-day variation in food consumption. In each visit, detailed food and beverage consumption was determined using a 24 h recall survey following the United States Department of Agriculture (USDA) five-step multiple-pass method [45]. Portion sizes were estimated using everyday utensils and a photographic album with pictures of local foods. A USDA composition table considering nutritional equivalencies was harmonized with local foods [46]. 

### 2.5. Estimation of Vitamin E and C Intake and Inadequacy 

Usual dietary intakes of vitamin E (as α-tocopherol equivalents) and vitamin C were calculated using the Multiple Source Method, a web-based tool developed by researchers at the European Prospective Investigation into Cancer and Nutrition (EPIC) to estimate the usual dietary intakes of nutrients and foods consumed by population individuals, available at http://msm.dife.de/ (accessed on 1 December 2017).

EAR in women for vitamin E (12 mg/day of α-tocopherol) and vitamin C (60 mg/day for women and 56 mg/day of vitamin C for 14–18-year-old-girls) were used to estimate the percentage of WRA with inadequate intakes using the following formula: [respondent’s current intake/EAR for the corresponding age category] × 100.

### 2.6. Food Sources

Regional foods with no exact equivalent available in the NDS-R database were divided into ingredients, and these were entered in the software as user recipes. Local teams were responsible for creating a recipe representing the same nutritional value as the original version and adjusted local food mandatory fortification. The data were obtained from national publications, cookbooks, and websites of each country and were contrasted with data from the R24H. Consistency checks were performed to minimize errors and verify key nutrient results. A detailed description of the adaptation of available food composition to a single database can be found in [46].

A total of 3351 types of food and beverages were reported in both R24H for all countries, grouped into ninety-three food items, according to nutritional similarities [47]. The items and foods were also categorized into eighteen food groups, representing a more extensive and more general group list. The main food groups providing vitamin E or C in each country were ranked by percent contribution to total dietary intake. 

### 2.7. Statistical Analyses 

Categorical variables are expressed as numbers of cases or percentages. The description of α-tocopherol and vitamin C intake is presented as mean, standard deviation and 10th , 50th, and 90th percentiles (in mg/day), and stratified by country, age (15–19, 20–34, 35–49 years old), SEL (high, medium, low), and nutritional status (underweight, normal weight, overweight, obese). The normality of variable distributions was evaluated using the Kolmogorov–Smirnov test. 

To compare the intake averages among different categories (by country, age group, SES, and nutritional status), one-way analysis of variance (ANOVA) and Bonferroni multiple comparisons post-test were used. Significance was established at a *p*-value < 0.05. Chi-square tests were performed to evaluate the association between categorical variables. For analysis of the homogeneity in variances, the Levene test was used. 

SPSS^®^ program (Statistical Package for the Social Sciences) version 24 for Windows was used for statistical analyses. Figures were designed using GraphPad Prism 5 for Windows (GraphPad Inc., San Diego, CA, USA), and tables were prepared using MS Excel 2017.

### 2.8. Ethics

The ELANS protocol was approved by the Western Institutional Review Board (#20140605) as well as the Ethics Review Boards of each participating institution and was registered at Clinical Trials (#NCT02226627). Written informed consent/assent was obtained from all study participants and confidentiality for the pooled data was maintained by using numeric identification codes instead of names. A secure file sharing system was used for data transfer.

## 3. Results 

### 3.1. Participant’s Characteristics 

The total sample size of this study included 3704 women. As shown in Table 1, the country with the largest number of participants was Brazil (*n* = 796, 21.6%), and the one with the smallest sample size was Costa Rica (*n* = 309, 8.3%). Most women included were adults, and only 14.6% were adolescents. The proportion of high SEL women was the lowest (14.3%), and women in medium and low SELs accounted for 43% and 42.7% of the sample, respectively. Regarding participants’ nutritional status, only 3.5% of women were underweight, and more than 57% had excess weight. 

### 3.2. Vitamin E Daily Intake and Assessment of Adequacy 

Dietary intake of vitamin E in WRA from ELANS are presented in Table 2 as mean ± standard deviation (S.D.) and 10th, 50th, and 90th percentiles by country, age group, SEL, and nutritional status. The average vitamin E intake in ELANS was 7.9 ± 3.2 mg/day (median 7.3 mg/day). Lower vitamin E consumption was observed in Brazil, Perú, Venezuela, and Chile. Ecuador and Argentina showed the highest intakes. Ecuador was the only country with a mean intake that reached the EAR levels.

Consumption was lower in older (35 to 49 years old) and overweight or obese WRA. Less intake was also detected in WRA of high or low SEL compared with medium level. At the 10th percentile, α-tocopherol intake was 4.6 mg/day (range from 3.6 mg/day in Brazil to 7.2 mg/day in Ecuador). At the 90th percentile, the mean intake of α-tocopherols in ELANS was 12 mg/day (range from 9.2 mg/day in Brazil to 17.6 mg/day in Ecuador). 

Vitamin E intake as a percentage of the EAR and as the proportion of participants consuming less than the EAR are presented in Table 3 and Figure 1. The mean intake of vitamin E in ELANS was 66% of the average requirement at a population level. This percentage was lower in Brazil, Venezuela, Chile, and Perú and higher in Argentina and Ecuador. Almost 90% of WRA from ELANS consumed vitamin E levels below the EAR (Figure 1). In all countries except Argentina, Ecuador, and Costa Rica, less than 6% of WRA consumed α-tocopherols above the EAR. 

### 3.3. Vitamin E Food Sources 

The top food sources of α-tocopherols in each country, ranked by percent contribution to total dietary intake, are shown in Figure 2. The primary sources of vitamin E in food consumed by ELANS participants were fats and oils. Their consumption accounted for 49% of vitamin E intake (range from 39% in Colombia to 63% in Ecuador). The types of vegetable oils most consumed by ELANS participants were sunflower, soybean, and corn oils. The content of vitamin E in sunflower oil is two to three times higher than in the other two oil types. Interestingly, the mean vitamin E intake in each country correlated positively with the % of sunflower oil consumed (*r* = 0.7213; *p* = 0.043) and negatively with the % of other oils consumed (*r* = −0.7201; *p* = 0.044). The second vitamin E source corresponded to vegetables, which provided 8% (6–12%) of this micronutrient. Together, oils and vegetables contributed to more than half of vitamin E provision in the eight countries analyzed. The remaining contribution was given by sweet baked products and snacks, followed by cereals, eggs, bread, red meat, and derived products as well as other foods, such as dairy, fruits, white meat, fish and seafood, legumes and nuts, non-alcoholic beverages, sweets and candy, and spices. 

### 3.4. Vitamin C Daily Intake and Assessment of Adequacy

Table 2 shows dietary vitamin C intake in ELANS by country, age group, SEL, and nutritional status as mean ± standard deviation (SD) and 10th, 50th, and 90th percentiles. The average intake of vitamin C was 94.4 ± 90.3 mg/day (median 72.9 mg/day). Seven countries (all except Argentina) showed mean intakes above the EAR of 60 mg/day. In Brazil and Ecuador, intakes doubled the EAR. High intake levels were also observed in Colombia and Perú. Costa Rica showed moderate consumption levels and Chile and Argentina were the countries with lower vitamin C intake. No differences in vitamin C consumption were observed by age or nutritional status. Lower intake was observed in medium and low SEL (*p* < 0.001).

Table 3 and Figure 1 show vitamin C intake as % of the EAR and the proportion of WRA consuming less than the EAR. The mean intake of vitamin C in ELANS was above the average requirement. However, 36% of all WRA from LatAm consumed less than the EAR. This percentage was highly variable: some countries showed high inadequacy, such as Argentina (78%) and Chile (59%), while others reported lower inadequacy, such as Ecuador (4%), Venezuela (16%), and Colombia (18%).

### 3.5. Vitamin C Food Sources 

Figure 2 shows the main food sources of vitamin C per country ranked by percent contribution to total dietary intake. Fruits and fruit juices accounted for at least 30% of the vitamin C provision in all the countries studied (mean of 46%; range from 32% (Chile) to 75% (Brazil)). The contribution of fruit versus juices as vitamin C sources differed among countries. In Brazil, Venezuela, Ecuador, and Perú, fruit juices were the main source of this vitamin, whereas in Chile, Argentina, and Costa Rica, juices contributed to less than 10% of the intake. Vegetables provided a mean of 23% of vitamin C [range from 10% (Brazil) to 32% (Costa Rica)]. Non-alcoholic beverages were also a significant vitamin C source in some countries, including Colombia (31%) as well as Chile and Argentina (20% in both countries). Cereals, snacks, and fast food as well as other sources contributed to the remaining vitamin C intake.

### 3.6. Combined Vitamin E and Vitamin C Inadequacy

Table 4 shows combined intakes of vitamin E and vitamin C in WRA from ELANS. Overall, one third of LatAm women consume less than the EAR for both vitamins. Adequate intake for both vitamins was detected in only 7.8% of WRA. Combined inadequate consumption was higher in Argentina (63.1%), Chile (57.7%), Costa Rica (42.4%), and Brazil (40.1%) and lower in Perú (20.2%), Colombia (18.3%), and Venezuela (16%). A very low vitamin E plus C inadequacy was observed in Ecuador (3.7%). 

## 4. Discussion

This study shows that most WRA in LatAm fail strongly to meet the current vitamin E intake recommendations, as previously reported in other populations [13]. Despite geographical distances and cultural differences from other countries worldwide, the magnitude of vitamin E inadequacies detected in LatAm were relatively similar to those previously reported in WRA from developed nations such as the US [14] and Spain [15]. Moreover, the median α-tocopherol intake of 7.9 mg/day in our study is somewhat higher than the global α-tocopherol intake of 6.2 mg/day reported in a recent systematic review, which analyzed 132 studies/46 countries, from 2000 to 2012 [13]. A “global” vitamin E inadequacy of 61% was calculated in that study, which is lower than 89.6% of WRA consuming less than the EAR in LatAm [13]. These differences are probably explained by the fact that most of the studies included in that systematic review were conducted in Europe, a region with low vitamin E inadequacy (55%), and only 3/34 studies were from Latin America (one from Mexico and two from Brazil). 

Our results show variable inadequacies in vitamin E intake among different LatAm countries from ELANS. Some, such as Brazil, Chile, Colombia, and Venezuela, showed extreme situations, with less than 3% of WRA consuming α-tocopherol above the EAR, similar to what has been reported in the US [14]. The only country showing mean intake levels above the EAR was Ecuador, although more than half of Ecuadorian WRA consumed less than the EAR. Globally, older women with excess weight consumed less vitamin E. This result could be pathophysiologically relevant because advanced maternal age [48] and obesity [49] independently increase the risk of adverse pregnancy outcomes, and the risk is even higher in aged women who are overweight [50]. However, this association was not consistent across all the ELANS countries and may be due to misreporting of energy intake. Indeed, underreporting in ELANS is significantly higher in women with excess weight (OR (95% CI): 1.91 (1.61–2.26) and 2.85 (2.40–3.38) in overweight and obese, respectively), and older age (OR (95% CI): 1.30 (1.04–1.62) in 20–34-year-old women and 1.39 (1.10–1.75) in 35–49-year-old women) [51]. The impact of misreporting may be especially relevant during the assessment of vitamin E intake because less energy coming from fats, which is a main food source of this micronutrient, than from other macronutrients was estimated in under-reporters [51]. Despite these considerations, vitamin E inadequacy was still high in young ELANS women with normal nutritional status. 

The cause and relevance of worldwide insufficient vitamin E intake are unclear. Usually, adults who consume strikingly low vitamin E from foods do not show deficient circulating levels [17,18]. The existence of adequate plasma vitamin E status concomitant with insufficient intake has been thoroughly discussed in expert panels and published reviews [9,17,18,37]. Underestimation of vitamin E intake through surveys may be explained, among other causes, by: (i) lack of specific food databases to be applied in different geographical settings; (ii) imprecise tabulation of oil types with variable vitamin E contents; (iii) uncertainties about fats or oils consumed in prepared foods where labels do not specify this information; and (iv) under-report of vitamin E intake from fatty foods. The need to review dietary reference intakes for vitamin E has also been discussed. The Panel on Dietetic Products, Nutrition and Allergies (NDA) from the European Food Safety Authority (EFSA) has concluded that EAR or Population Reference Intakes (PRIs) for vitamin E (as α-tocopherol) cannot be derived for adults due to lack of information [34]. Despite these limitations, EFSA sets an adequate intake (A.I.) of 11 mg/day of vitamin E in adult women [35]. The recent Dietary Guidelines for the Americans from the US government recommend adequate vitamin E intakes to be met primarily through natural foods [52]. Indeed, both dietary and fortified food-interventions have shown to be useful as means to improve vitamin E status in women [53,54]. Supplements are also considered as options when needs cannot be met otherwise [52]. 

The main food sources of vitamin E in ELANS are similar to those observed in the Spanish ANIBES (Anthropometry, Intake and Energy Balance in Spain) study, which identified oils and vegetables as the food sources providing 46% and 11% of vitamin E intake in Spanish adults, respectively. By contrast, the top vitamin E sources consumed by adults from the US are cereals and baked sweets (cakes, cookies, pies, and doughnuts), and oils and vegetables only account for 7% and 4% of its provision, respectively [14]. The fact that synthetic vitamin E in fortified foods (e.g., fortified ready-to-eat cereals) are also registered in US surveys probably explains this difference between countries. The provision of vitamin E from supplements in LatAm WRA from ELANS is unknown because this survey only registered vitamin C and E intake derived from food. As far as we are aware of, there is little information available on vitamin supplement use in WRA from LatAm. In Chile, the last Chilean National Health Survey (2016–2017) showed a very low consumption of supplements providing folic acid in WRA [55]. Thus, the intake of supplemental vitamin E is also probably small.

As described above, in LatAm and other western countries, a high proportion of dietary vitamin E is provided by fats and oils, together with sweet baked products, none of which are rich in α-tocopherol but contribute to α-tocopherol intake because of their high consumption. In agreement, a recent report showed that Latin American women of childbearing age consumed diets with low diversity of nutrients, characterized by high intake of cereals and low consumption of vitamin E-rich foods such as green leafy vegetables (6.8%) and nuts or seeds (2.8%) [56]. Dietary diversity usually reflects the high quality of the diet and is associated with an increased probability of adequate micronutrient intake [57]. Promoting intake of vitamin E-rich foods and dietary oils containing high vitamin E content, i.e., sunflower oil, may serve to reduce vitamin E inadequacies in LatAm WRA. 

Vitamin C intake is higher than vitamin E consumption in the US [16] and most European countries [15,58]. As vitamin C may help recycle the antioxidant function and preserve the stores of functional vitamin E, its intake has also been proposed to explain the low incidence of overt vitamin E deficiency in populations with very low tocopherol intake. Although the mean vitamin C intake in LatAm is above the EAR, its consumption is highly variable among different countries. It also seems to depend on the SEL, with lower intakes in lower socioeconomic categories. Vitamin C deficiency in low-income populations has also been pointed out in the US [59]. Furthermore, the EAR for smoking women is higher (95 mg/day) than for non-smokers [12]. Smoking was not registered in ELANS; thus, we could not calculate the adequacy of vitamin E in LatAm WRA smokers. Consumption of fruit juices, fruits, and vegetables accounted for around half of the vitamin C intake in the ELANS sample altogether and in each country individually, as observed previously in other countries [15,16,59]. 

Despite existing evidence showing that vitamin E-deficient diets are associated with infertility and adverse pregnancy outcomes in animals [26,27,28,29] and humans [30,31,32], α-tocopherol status is not frequently evaluated in infertile women or during pregnancy. Because of increased placental, embryonic, and fetal nutritional demands of macro and micronutrients during gestation, subclinical inadequacy or insufficient intake in WRA with normal plasma vitamin E status may still have adverse effects. This situation may be especially relevant in WRA with increased oxidative status, such as smokers, metabolically compromised, or aged women. Clinical studies specially designed to evaluate the effect of dietary interventions or supplementation with vitamin E on WRA on vitamin E status, oxidative damage biomarkers, and fertility or pregnancy outcomes could test this hypothesis.

This study provides novel information on vitamin E and C intake in nationally representative samples of WRA in urban populations from eight Latin American countries. The distribution of nutritional status categories is comparable to reported estimates for urban LatAm women recently published by the United Nations Food and Agricultural Organization [60]. One strength of this report is the use of quantitative data analyzed through standardized methods with simultaneous data collection, which allows for better and more reliable comparisons among countries. However, this study has several limitations. First, the inherent bias when collecting dietary intake data, especially using 24 h recalls, has to be considered. Second, nutrient contents of foods and dishes were estimated using US food composition databases. Although these were modified according to local recipes, modifications may have also resulted in additional error sources. In addition, our data are limited to urban populations but are not representative of rural and overall LatAm populations. 

## 5. Conclusions

Our study determined a high inadequacy for vitamin E and a lower yet moderate inadequacy for vitamin C in LatAm WRA. A combined deficient intake of both vitamins was observed in one-third of ELANS participants. The implications of low worldwide consumption of antioxidant vitamins are still unclear. Further research on this subject is required to evaluate the implications of these inadequacies on fertility and pregnancy, especially in women of reproductive age with increased oxidative status, such as smokers, metabolically compromised, or aged women. Prospective clinical studies designed to evaluate the potential adverse effects of well-defined vitamin E forms plus other nutritional co-antioxidants on biomarkers of oxidative damage and their association with adverse outcomes of fertility and pregnancy are necessary. In the meantime, evaluating and implementing strategies to promote the production and intake of vitamin E-rich foods may help to prevent potential unknown consequences of insufficient consumption of this micronutrient.

## Figures and Tables

**Figure 1 nutrients-13-01954-f001:**
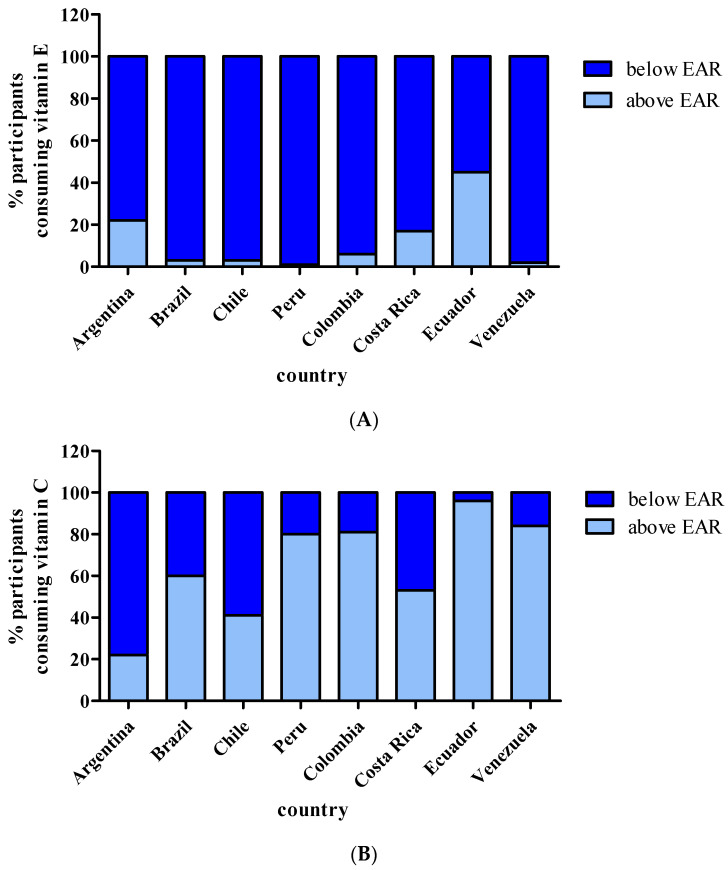
Prevalence of adequacy (% of population above EAR) and inadequacy (% of population below EAR) for vitamins E (**A**) and C (**B**) in LatAm WRA participating in ELANS.

**Figure 2 nutrients-13-01954-f002:**
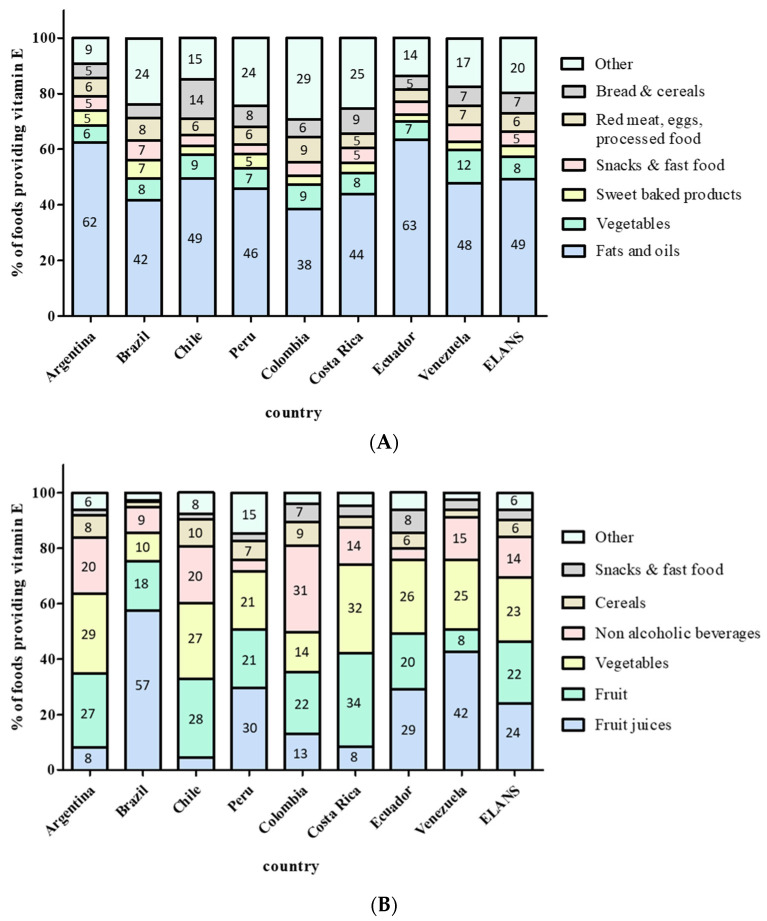
Dietary contributions of food sources to vitamin E (**A**) and vitamin C (**B**) intakes in LatAm WRA participating in ELANS.

**Table 1 nutrients-13-01954-t001:** Sample characteristics: distribution of 3704 15–49-year-old WRA participants of ELANS in different countries and sociodemographic categories.

Variable	Number (*n*)	Frequency (%)
Country		
Argentina	521	14.1
Brazil	798	21.5
Chile	345	9.3
Perú	480	13.0
Colombia	464	12.5
Costa Rica	309	8.3
Ecuador	324	8.7
Venezuela	463	12.5
Age (years old)		
15 to 19	539	14.6
20 to 34	1771	47.8
35 to 49	1394	37.6
Socioeconomic level (SEL)		
High	529	14.3
Medium	1593	43.0
Low	1582	42.7
Nutritional status		
Underweight	128	3.5
Normal weight	1444	39.0
Overweight	1177	31.8
Obesity	953	25.7

**Table 2 nutrients-13-01954-t002:** Vitamin E and C daily intake in WRA participants of ELANS in different countries and groups.

		Vitamin E (mg/day)					Vitamin C (mg/day)				
		Mean ± SD.	*p*	Percentiles			Mean ± SD.	*p*	Percentiles		
				10th	50th	90th			10th	50th	90th
ELANS total sample		7.9 ± 3.2	*-*	4.6	7.3	12.1	94.4 ± 90.3	*-*	31.2	72.9	167.5
Country	Argentina	9.9 ± 2.9 ^a^	<0.001	6.3	9.7	14.0	46.6 ± 23.2 ^a^	<0.001	24.5	41.5	73.9
	Brazil	6.2 ± 2.3 ^b^		3.6	5.8	9.2	120.8 ± 163.3 ^b^		24.6	72.9	252.3
	Chile	6.9 ± 2.4 ^c^		4.4	6.5	9.6	60.5 ± 33.7 ^a,c^		26.2	51.4	105.1
	Perú	6.9 ± 1.9 ^c^		4.8	6.6	9.2	98.8 ± 57.6 ^d^		46.8	89.0	159.1
	Colombia	7.9 ± 2.6 ^d^		5.1	7.7	11.0	100.7 ± 48.2 ^d^		48.5	94.0	163.6
	Costa Rica	9.0 ± 2.7 ^e^		5.8	8.6	12.8	71.9 ± 45.4 ^c^		28.2	61.2	126.4
	Ecuador	12.1 ± 4.2 ^f^		7.2	11.5	17.6	127.1 ± 53.9 ^b^		69.3	118.5	190.1
	Venezuela	6.9 ± 2.2 ^c^		4.3	6.7	9.6	109.6 ± 51.7 ^b,d^		47.7	103.2	180.7
Age (years old)	15 to 19	8.4 ± 3.3 ^a^	<0.001	4.9	7.7	12.9	91.2 ± 70.8	0.645	30.2	71.5	170.0
	20 to 34	8.0 ± 3.2 ^b^		4.6	7.4	12.0	95.4 ± 92.8		31.8	77.1	166.1
	35 to 49	7.6 ± 3.1 ^c^		4.4	7.1	11.9	94.5 ± 93.7		30.9	72.6	168.2
SEL	High	7.4 ± 3.1 ^a^	<0.001	4.3	6.9	11.5	114.1 ± 93.9 ^a^	<0.001	34.8	90.5	204.2
	Medium	8.2 ± 3.4 ^b^		4.6	7.7	12.7	96.6 ± 110 ^b^		29.3	72.8	168.9
	Low	7.8 ± 2.9 ^a^		4.5	7.2	11.7	85.7 ± 61.1 ^c^		31.8	72.5	152.1
Nutritional status	Underweight	8.7 ± 3.3	0.004	5.2	8.0	13.4	90.7 ± 64.9	0.935	31.1	70.9	174.7
	Normal weight	8.0 ± 3.1		4.6	7.5	12.2	94.2 ± 82.3		30.9	74.7	170.9
	Overweight	7.8 ± 3.2		4.5	7.2	12.0	95.5 ± 84.8		32.6	79.6	163.2
	Obese	7.8 ± 3.1		4.4	7.2	11.9	94.0 ± 109.5		30.1	70.2	162.9

Different letters represent groups showing statistically significant differences (*p* < 0.05). SEL: socioeconomic level.

**Table 3 nutrients-13-01954-t003:** Vitamin E and C intake (% of EAR and % below EAR) by country, age, SEL, and nutritional status.

			Vitamin E			Vitamin C		
			% of EAR		Below 100%	% of EAR		Below 100%
			Mean ± SD	*p*		Mean ± SD	*p*	
ELANS total sample		3704	66.0 ± 26.4		89.6%	158.6 ± 151.2		36.3%
Country	Argentina	521	82.3 ± 24.5 ^a^	<0.001	77.9%	78.1 ± 38.7 ^a^	<0.001	78.1%
	Brazil	798	51.7 ± 19.3 ^b^		97.5%	202.4 ± 272.8 ^b^		40.1%
	Chile	345	57.3 ± 19.7 ^c^		97.1%	101.8 ± 57.1 ^a,d^		58.8%
	Perú	480	57.7 ± 15.4 ^c^		98.8%	166.3 ± 98.1 ^c^		20.2%
	Colombia	464	65.8 ± 21.5 ^d^		94.4%	169.1 ± 81 ^c^		18.5%
	Costa Rica	309	74.9 ± 22.5 ^e^		82.8%	120.7 ± 76 ^d^		46.9%
	Ecuador	324	100.5 ± 34.9 ^f^		55.2%	213.8 ± 90.6 ^b^		4.0%
	Venezuela	463	57.2 ± 18.5 ^c^		97.6%	184.4 ± 87.3 ^b,c^		16.0%
Age (years old)	15 to 19	539	69.9 ± 27.5 ^a^	<0.001	86.1%	160.4 ± 124.7	0.921	36.0%
	20 to 34	1771	66.5 ± 26.5 ^b^		90.0%	159.0 ± 154.6		35.2%
	35 to 49	1394	63.7 ± 25.7 ^c^		90.4%	157.5 ± 156.2		37.8%
SEL	High	529	61.7 ± 26.1 ^a^	<0.001	92.6%	191.4 ± 157.4 ^a^	<0.001	27.8%
	Medium	1593	68.5 ± 28 ^b^		87.1%	162.3 ± 184.2 ^b^		38.0%
	Low	1582	64.8 ± 24.5 ^a^		91.1%	144.0 ± 102.5 ^c^		37.5%
Nutritional status	Underweight	128	72.7 ± 27.4 ^a^	0.004	83.6%	154.4 ± 110.7	0.960	39.8%
	Normal weight	1444	66.9 ± 26.1 ^ab^		89.3%	158.9 ± 138.6		37.1%
	Overweight	1177	64.8 ± 26.8 ^b^		90.0%	160.0 ± 141.7		31.0%
	Obese	953	65.1 ± 26.2 ^b^		90.2%	157.1 ± 182.7		41.2%

Different letters represent groups showing statistically significant differences (*p* < 0.05). EAR: estimated average requirement; SEL: socioeconomic level.

**Table 4 nutrients-13-01954-t004:** Vitamin E and C combined adequacies and inadequacies in different countries from ELANS.

	Inadequacy of Vitamin E EAR		Adequacy of Vitamin E EAR	
Country	Inadequacy of Vitamin C EAR	Adequacy of Vitamin C EAR	Inadequacy of Vitamin C EAR	Adequacy of Vitamin C EAR
Argentina	63.1%	14.8%	15.0%	7.1%
Brazil	40.1%	57.4%	0.0%	2.5%
Chile	57.7%	39.4%	1.2%	1.7%
Perú	20.2%	78.5%	0.0%	1.3%
Colombia	18.3%	76.1%	0.2%	5.4%
Costa Rica	42.4%	40.5%	4.5%	12.6%
Ecuador	3.7%	51.5%	0.3%	44.4%
Venezuela	16.0%	81.6%	0.0%	2.4%
ELANS total sample	33.7%	55.9%	2.6%	7.8%

## Data Availability

Due to ethical and legal restrictions of the eight institutions involved, the data underlying this study are available upon request and must be approved by the Publishing Committee of ELANS. Data are available from the ILSI Argentina Institutional Data Access for researchers who meet the criteria for access to confidential data and after approval by the Publishing Committee of ELANS. To apply for access to these data, interested researchers must submit a detailed project proposal to the above Institution. The authors confirm that the data underlying this study will be shared provided that requests are submitted through appropriate channels. Requests for the data can be made to the Executive Secretary of the Institution, Mrs Patricia Torres: ptorres@ilsi.org.ar.

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
