# Peer review of "Intake of Vitamin E and C in Women of Reproductive Age: Results from the Latin American Study of Nutrition and Health (ELANS)"

_nutrients, 2021, doi:10.3390/nu13061954_

Round 1

Reviewer 1 Report

I have reviewed the manuscript “Intake of Vitamin E and C in Women of Reproductive Age: Results from the Latin American Study of Nutrition and Health”. The study presents a large volume of valuable data to investigate vitamin E and C adequacy among women of reproductive age in Latin American countries. The study aim is relevant, the manuscript is very-well written, organized and easy to follow. Major limitations of the study are acknowledged. Please see below a few comments/suggestions to review the manuscript.

-The last sentence of the abstract does not seem suitable to me. Although this statement is most likely correct, I do not think this is something that can actually be learnt by reading this study.

-how does the distribution of nutritional status compares with reported estimations of these countries? Are we looking at a representative population?

-how do you explain that underweight WRA had the highest vitamin E intake?

-I am missing part B of figure 2.

-please elaborate on future perspectives. What steps are to be taken now? What has been done in other regions of the world and could be implemented in LatAm?

-number of subject is not correct in line 119 (?)

Author Response

Santiago, May 27th, 2021

Dear Reviewer 1:

We were pleased to receive your review on our manuscript “Intake of Vitamin E and C in Women of Reproductive Age: Results from the Latin American Study of Nutrition and Health (ELANS)”, which was also positively evaluated by reviewer 2. Your comments and suggestions were very constructive and helpful and certainly aided in improving our manuscript. Please find the answers to your observations below, after each of your comments, in italics:

I have reviewed the manuscript “Intake of Vitamin E and C in Women of Reproductive Age: Results from the Latin American Study of Nutrition and Health”. The study presents a large volume of valuable data to investigate vitamin E and C adequacy among women of reproductive age in Latin American countries. The study aim is relevant, the manuscript is very-well written, organized and easy to follow. Major limitations of the study are acknowledged. Please see below a few comments/suggestions to review the manuscript.

Thank you for your positive and constructive comments on our manuscript. We have prepared a revised version of the document. Our answers are included below each observation, in italics.

 -The last sentence of the abstract does not seem suitable to me. Although this statement is most likely correct, I do not think this is something that can actually be learnt by reading this study.

 We agree that the last sentence of our abstract was somehow disconnected from our actual work. We rephrased the end of the paragraph to integrate the results of our study with information that we would like to keep in this section: i) the lack of certainty on the relevance of low vitamin E (and C) consumption and ii) the need for further research. We hope you find an improvement in this version of the abstract (lines 39-41).

-how does the distribution of nutritional status compare with reported estimations of these countries? Are we looking at a representative population?

 Thank you for pointing out this observation. The distribution of nutritional status categories at a regional level is comparable to recent FAO estimates for LatAm. We did not evaluate nutritional status categories in each country in this paper. Individual information of anthropometric variables of ELANS WRA, in addition to other sociodemographic characteristics, have been recently reported for the 8 ELANS countries individually. We have included this information and the corresponding references in lines 156-157 and 413-415.

 -how do you explain that underweight WRA had the highest vitamin E intake?

 We agree that this result is intriguing. However, although the mean vitamin E intake in underweight women was near 10% higher than in women with normal weight, it must be pointed out that differences in vitamin E intake among nutritional status categories were not significant (as determined by Bonferroni post-test). One possible explanation for this result is a higher intake of vitamin E-rich foods in underweight WRA from high SEL. Indeed, an interaction between SEL and nutritional status revealed that underweighted and normal-weight women from higher SEL in some LatAm countries (e.g., Peru and Ecuador) consumed more diverse diets than overweight and obese women from low and middle SEL [1]. Another plausible explanation is underreporting of fat consumption in normal and overweight/obese women, as discussed in lines 338-341 of the revised manuscript.

 Gómez, G.; Nogueira Previdelli, Á.; Fisberg, R.M.; Kovalskys, I.; Fisberg, M.; Herrera-Cuenca, M.; Cortés Sanabria, L.Y.; Yépez García, M.C.; Rigotti, A.; Liria-Domínguez, M.R., et al. Dietary Diversity and Micronutrients Adequacy in Women of Childbearing Age: Results from ELANS Study. Nutrients 2020, 12, doi:10.3390/nu12071994.

-I am missing part B of figure 2.

We are very sorry that Fig2B was not included in the editorial version of our manuscript used for the review. We have re-inserted it in the revised version.

-please elaborate on future perspectives. What steps are to be taken now? What has been done in other regions of the world and could be implemented in LatAm?

We agree that the original version of our manuscript lacked sufficient information on perspectives and further steps regarding vitamin E inadequacies in WRA. We have elaborated on these important issues on lines 357-361, 384-386 and 407-409 of the discussion section. We also included a paragraph with conclusions where we comment on these points too.

-number of subjects is not correct in line 119 (?)

This mistake was corrected; thank you for noticing it.

I also would like to let you know that we added a small yet interesting new result in our manuscript. We thought about and performed this analysis during the preparation of this version and find that it adds to the results on food sources. The following text was added: “The types of vegetable oils most consumed by ELANS participants were sunflower, soybean and corn oils. The content of vitamin E in sunflower oil is two to three times higher than in the other two oil types. Interestingly, the mean vitamin E intake in each country correlated positively with the % of sunflower oil consumed (r=0.7213; p=0.043) and negatively with the % of other oils consumed (r= -0,7201; p=0.044).” (lines 255-260). I hope you agree with including this additional result.

I look forward to receiving your comments on the revised manuscript.

Best regards,

Dolores Busso

                                                                                    Associate Professor

                                                                                    School of Medicine

                                                                                    Universidad de los Andes, Chile

Reviewer 2 Report

An interesting original study about vitamin E nad C intake in   the Latin American Study of Nutrition and Health, showing high deficency for vitamin E intake and moderate, inadequacy for vitamin C  intake in women in reproductive age.

A combined deficient intake of 
both vitamins was observed in one-third of participants.

Only minor queries:

Page 2 line 59 you should add: "The antioxidant activity of vitamin E is strongly dependent on the action of other biological agents, such as ascorbic acid, vitamin B3, selenium and glutathione. " and cite an artyicle such as: doi: 10.1007/s13668-020-00322-4.

Figure 1 should be grafically updated.

Discussion should be divided in a discussion and conclusion paragraph, where future developments following this article must be better assessed.

Thank You

Author Response

Santiago, May 27th, 2021

Dear Reviewer 2:

Thank you for reviewing our manuscript "Intake of Vitamin E and C in Women of Reproductive Age: Results from the Latin American Study of Nutrition and Health (ELANS)".  We were glad to know that it was positively evaluated by both assigned reviewers. Your constructive observations certainly aided in improving our manuscript. Please find the answers to your observations below, after each of your comments, in italics:

REVIEWER 2:

An interesting original study about vitamin E and C intake in the Latin American Study of Nutrition and Health, showing high deficiency for vitamin E intake and moderate, inadequacy for vitamin C intake in women in reproductive age. A combined deficient intake of both vitamins was observed in one-third of participants.

 Thank you for your comments and suggestions, which have indeed improved our manuscript. Please find our answers below, in italics.

 Only minor queries:

Page 2 line 59 you should add: "The antioxidant activity of vitamin E is strongly dependent on the action of other biological agents, such as ascorbic acid, vitamin B3, selenium and glutathione. " and cite an article such as: doi: 10.1007/s13668-020-00322-4.

As suggested, we included information that points out the interaction of vitamin E with other antioxidant nutrients (lines 60-62 and 433-434).

 Figure 1 should be grafically updated.

 Unfortunately, I do not understand this observation. I am sorry about that. Figure 2B was not included in the editorial version of our manuscript, and we have re-inserted it in the revised version. Did you intend to comment on Figure 2?

Discussion should be divided in a discussion and conclusion paragraph, where future developments following this article must be better assessed.

Following your recommendation, we prepared a Discussion section where we summarize the main finding, describe the implications of this information, and propose future steps on research in this area. I believe this modification improved the manuscript as it allows rounding up our contribution better (lines 425-438)

I also would like to let you know that we added a small yet interesting new result in our manuscript. We thought about and performed this analysis during the preparation of this version and find that it adds to the results on food sources. The following text was added: “The types of vegetable oils most consumed by ELANS participants were sunflower, soybean and corn oils. The content of vitamin E in sunflower oil is two to three times higher than in the other two oil types. Interestingly, the mean vitamin E intake in each country correlated positively with the % of sunflower oil consumed (r=0.7213; p=0.043) and negatively with the % of other oils consumed (r= -0,7201; p=0.044).” (lines 255-260). I hope you agree with including this additional result.

I look forward to receiving your comments on the revised manuscript.

Best regards,

Dolores Busso

                                                                                    Associate Professor

                                                                                    School of Medicine

                                                                                    Universidad de los Andes, Chile
